# Innervation of supraclavicular adipose tissue: A human cadaveric study

Will Sievers[1]*, Joseph A. Rathner[1,2], Rodney A. Green[1], Christine Kettle[1], Helen R. Irving[1], Donna R. Whelan[1], Richard G. D. Fernandez[3], Anita Zacharias[1]

**1** Department of Pharmacy and Biomedical Sciences, La Trobe Institute for Molecular Science, La Trobe University, Bendigo, Victoria, Australia, **2** Department of Physiology, School of Biomedical Sciences University of Melbourne, Parkville, Victoria, Australia, **3** Department of Physiology Anatomy and Microbiology, School of Life Sciences, La Trobe University, Melbourne, Victoria, Australia

* W.Sievers@latrobe.edu.au

**Data Availability Statement:** All relevant data are within the paper and its Supporting Information files.

## Abstract

Functional brown adipose tissue (BAT) was identified in adult humans only in 2007 with the use of fluorodeoxyglucose positron emission tomography imaging. Previous studies have demonstrated a negative correlation between obesity and BAT presence in humans. It is proposed that BAT possesses the capacity to increase metabolism and aid weight loss. In rodents it is well established that BAT is stimulated by the sympathetic nervous system with the interscapular BAT being innervated via branches of intercostal nerves. Whilst there is evidence to suggest that BAT possesses beta-3 adrenoceptors, no studies have identified the specific nerve branch that carries sympathetic innervation to BAT in humans. The aim of this study was to identify and trace the peripheral nerve or nerves that innervate human BAT in the supraclavicular region. The posterior triangle region of the neck of cadaveric specimens were dissected in order to identify any peripheral nerve branches piercing and/or terminating in supraclavicular BAT. A previously undescribed branch of the cervical plexus terminating in a supraclavicular adipose depot was identified in all specimens. This was typically an independent branch of the plexus, from the third cervical spinal nerve, but in one specimen was a branch of the supraclavicular nerve. Histological analysis revealed the supraclavicular adipose depot contained tyrosine hydroxylase immunoreactive structures, which likely represent sympathetic axons. This is the first study that identifies a nerve branch to supraclavicular BAT-like tissue. This finding opens new avenues for the investigation of neural regulation of fat metabolism in humans.

## Introduction

White adipose tissue (WAT) makes up the majority of fat in the human body and is primarily used to store energy in the form of triacylglycerols. Extensive research has been conducted to better understand and control WAT, particularly in relation to human metabolism, obesity, and obesity related diseases [1, 2]. There is growing emphasis on improving current weight loss strategies and therapies as the prevalence of overweight and obesity in adults increases. Globally, in 2016, 52% of adults were overweight or obese [3]. Additionally, overweight and

**Funding:** Dr. Whelan is the recipient of an Australian Research Council Australian Discovery Early Career Research Award (DE200100584) funded by the Australian Government. The Australian Government had no role in study design, data collection and analysis, decision to publish, or preparation of the manuscript.

**Competing interests:** The authors have declared that no competing interests exist.

obese individuals have an increased risk of comorbidities such as diabetes, cardiovascular disease and some cancers [4]. Shortcomings associated with the current weight loss strategy of caloric restriction are due to slowing of resting metabolic rate [5, 6]. Beyond lifestyle intervention, bariatric surgery is invasive and presents risks of complications such as pulmonary embolism, enteric leak or gastrointestinal bleeding [7]. While pharmacological intervention is also an option, these drugs (i.e. glucagon-like peptide-1 receptor agonists, or adrenergic receptor agonists) typically have gastrointestinal or cardiovascular side effects [8, 9].

In contrast to WAT, brown adipose tissue (BAT), which makes up only a fraction of total fat mass in mammals, and has historically been under-studied, presents a relatively new target for novel obesity treatments because BAT has the capacity to produce heat (thermogenesis) [10]. By specifically activating BAT, this unique ability could be harnessed to expend energy that might otherwise be conserved resulting in an increased metabolic rate with potential for applications in weight loss therapies. While active BAT has been reported in obese individuals [11], studies have consistently observed in humans that obesity is inversely correlated with BAT activity [2, 12, 13] and BAT volume [14].

During thermogenesis, BAT expends energy in a futile cycle, where mitochondrial activity is "wasted" as heat, rather than producing ATP. This can manifest in two ways: BAT can help maintain optimum body temperature in cold conditions, or increase resting metabolic rate in order to maintain a healthy body-weight [15–17]. Metabolically active BAT has been identified in adult humans using fluorodeoxyglucose positron emission tomography (FDG PET) imaging [18]. Specifically, the supraclavicular region, located behind the clavicle, has been shown to possess major depots of BAT [19, 20] that are the most metabolically active [21], making it an ideal target for further study (Fig 1).

In rodents it has previously been shown that the sympathetic nervous system controls activation of BAT [22–24]. Sympathetic neurons release varying amounts of norepinephrine at their junction with BAT in order to tightly regulate the activity of the tissue. The nerves that synapse with BAT are specific, and as such, provide very targeted activation of BAT. The peripheral nerves carrying the sympathetic fibres that innervate interscapular BAT in rats are branches from up to five separate intercostal nerves (thoracic ventral rami) arising from directly beneath the interscapular (between the shoulder blades) depot of BAT [25]. These nerves are commonly utilised for *in vivo* nerve recording experiments as a means of assessing sympathetic drive to the interscapular depot of BAT in rats [26]. Despite the evidence in rats of sympathetic innervation to interscapular BAT, it remains to be demonstrated if the sympathetic nervous system innervates BAT in humans.

There is evidence to suggest that human BAT expresses adrenoceptors [18, 27], and possesses norepinephrine transporters [28, 29], providing some indication that human BAT is under sympathetic regulation. However, no specific peripheral nerve responsible for carrying innervation to BAT has been identified to date. Major differences in the location of BAT between rodents and humans have previously been identified. Rodents possess a major BAT depot in the interscapular region [10] whereas adult humans possess large depots of BAT in the supraclavicular, paraspinal and abdominal regions [14, 30].

Therefore, we investigated the innervation of supraclavicular adipose depots in human cadaveric specimens. These adipose depots lie deep in the region described anatomically as the posterior triangle of the neck (Fig 1). The main nerves innervating structures in the posterior triangle are branches of the cervical plexus (formed by ventral rami of cervical spinal nerves) with both superficial (e.g., supraclavicular nerve) and deep branches (e.g., dorsal scapular nerve) [31]. Since BAT is found in the supraclavicular region [30], we reasoned that BAT may form a separate or intermingled part of the adipose depot found in this region. We therefore hypothesised that the supraclavicular depots of adipose tissue may be innervated by a branch or branches of the cervical

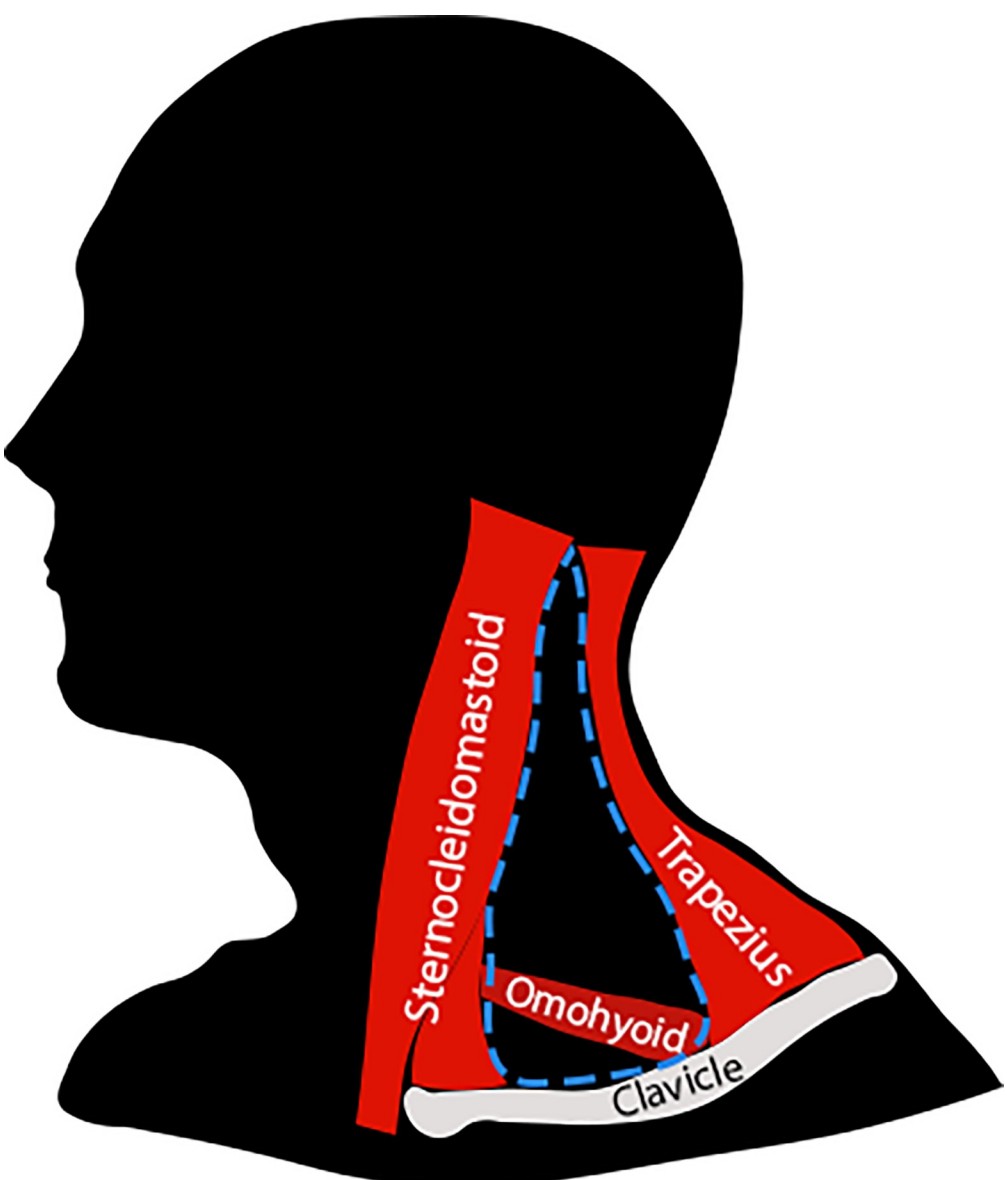

**Fig 1. Diagram outlining the posterior triangle of the neck.** The posterior triangle of the neck is shown by the blue outline. The adipose depot of the supraclavicular region is located in the inferior (lower) portion of the posterior triangle. The origin of the cervical plexus and its branches mainly lie in the superior (upper) part of the posterior triangle.

plexus. We identified and traced peripheral nerves to this adipose depot in the supraclavicular region. Morphological assessment of the adipose tissue revealed BAT-like features. Moreover, tyrosine hydroxylase immunoreactive structures were identified within the BAT-like adipose tissue collected from the supraclavicular region, suggestive of sympathetic innervation.

## Results

### Anatomical investigation of pattern of innervation

In order to investigate the innervation of adipose tissue within the supraclavicular region, five dissections of three adult cadavers were performed. A sixth dissection could not be

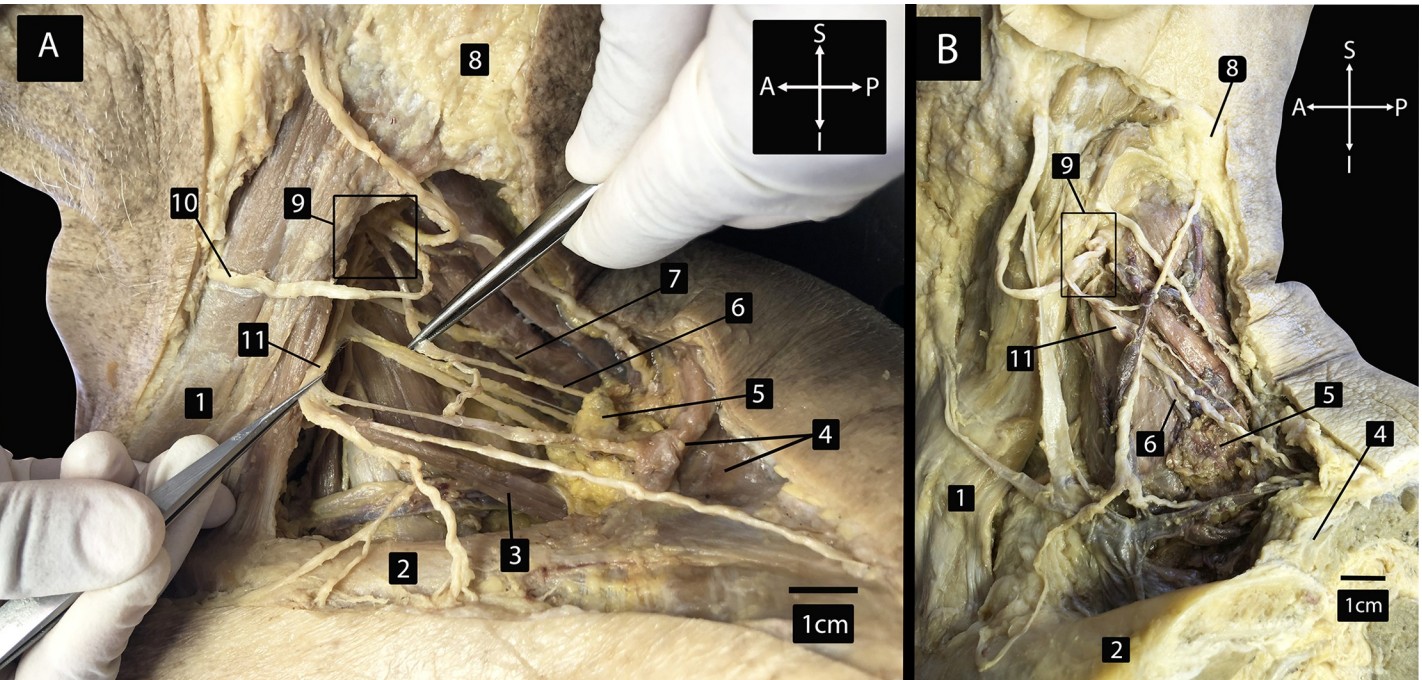

**Fig 2. Lateral view of left side of dissected specimens.** Dissected specimens showing depot of supraclavicular adipose tissue and surrounding nerves in the posterior triangle of the neck. (A) Specimen number 2. Innervation to adipose tissue as direct branch from cervical plexus. (B) Specimen number 1. Innervation to adipose tissue as branch of supraclavicular nerve. 1 = sternocleidomastoid muscle; 2 = clavicle (cut in B); 3 = omohyoid muscle (A); 4 = trapezius muscle; 5 = supraclavicular depot of adipose tissue; 6 = nerve innervating supraclavicular adipose tissue; 7 = dorsal scapular nerve (A); 8 = subcutaneous white adipose tissue (A); 9 = Branches of cervical plexus marked by the box; 10 = transverse cutaneous nerve of the neck (A); 11 = supraclavicular nerve; A = anterior; P = posterior; S = superior; I = inferior.

performed as the right side of specimen 1 had already been dissected for teaching purposes. To begin the dissection, the region of the posterior triangle was identified by locating the sternocleidomastoid muscle, collarbone (clavicle) and trapezius muscle (Fig 1). The skin, muscle and connective tissue overlying the posterior triangle were removed retaining all nerve branches of the cervical plexus (Fig 2). A depot of adipose tissue in the supraclavicular region within the posterior triangle, a location consistent with previous functional PET scan studies [27], was tentatively identified as BAT based on coloration [32]. This is clearly shown in Fig 2B by comparing WAT (structure 8) with supraclavicular depot of adipose tissue (structure 5). All previously described branches of the cervical plexus were identified and branches closely related to the supraclavicular depot of adipose tissue were isolated and followed to identify specific branches that terminated in adipose tissue. In all five dissections, branches of the cervical plexus were identified in the posterior triangle, including the relatively superficial supraclavicular branches that have been previously described as innervating skin over the posterior triangle and more inferiorly onto the chest and shoulder regions. A single nerve within each of the three specimens was found to terminate in the visibly darker adipose tissue, and therefore possibly BAT. Interestingly, two distinct patterns of innervation to this adipose tissue were identified (Table 1). In four of the five dissections, the nerve emerged as an independent branch of the cervical plexus from the ventral ramus of third cervical spinal nerve (Fig 2A). In the remaining dissection, the nerve emerged as a branch of the supraclavicular nerve (Fig 2B). This marks the first time innervation to supraclavicular adipose tissue has been reported in humans.

**Table 1. Details of cadaver specimens.**

| Cadaver specimen number | Age | Sex | Side of investigation | Branch of cervical plexus |
|---|---|---|---|---|
| 1 | 83 | M | Left | Branch of supraclavicular nerve |
| 2 | 84 | M | Right | Ventral ramus of third cervical spinal nerve |
| 2 | 84 | M | Left | Ventral ramus of third cervical spinal nerve |
| 3 | 72 | F | Right | Ventral ramus of third cervical spinal nerve |
| 3 | 72 | F | Left | Ventral ramus of third cervical spinal nerve |

### Histological characterization of adipose tissue and innervation

To determine the nature of the darker adipose tissue from the supraclavicular region, we extracted adipose tissue for further histological analysis. BAT is typically described as consisting of smaller adipocytes with multi-locular intracellular lipid droplets [33]. This contrasts with WAT which consists of larger adipocytes with only a single large lipid droplet [33, 34]. However, BAT can appear with uni-locular fat droplets, particularly in humans [35]. Additionally, in humans BAT typically appears interspersed between white adipocytes, rather than large depots of exclusively brown adipocytes [35]. Cell size has been used previously to identify distinct populations of adipocytes [33, 35].

To quantify differences between typical WAT and the presumed BAT depot, we examined cell morphology. Samples of the presumed BAT from the supraclavicular region were compared with samples of WAT taken from the armpit (axilla) of the same cadaveric specimen. Samples from a single specimen were used as studies have reported that intra-individual site-to-site variability in adipocyte size is greater than variability of the same site between individuals [36–38]. The WAT samples consist of typically large adipocytes with single large lipid droplets with a few intermingling smaller adipocytes. The darker adipocytes from the supraclavicular region contain some large cells and multiple cells that are generally smaller cells with a more variegated appearance to the tissue (Fig 3). Quantitative analysis of cell morphology demonstrated that adipocytes taken from the supraclavicular region are significantly smaller ($1268 \pm 60 \ \mu m^2$) than adipocytes from the axilla ($2812 \pm 135 \ \mu m^2$) (Fig 3). The comparatively smaller cell size supports the contention that the supraclavicular adipocytes are BAT-like [35].

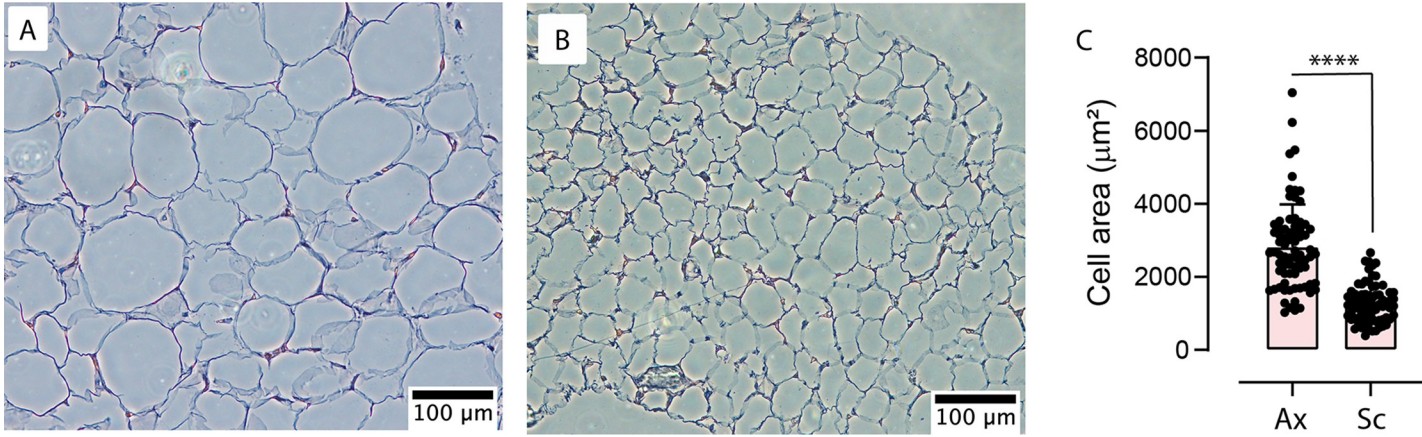

**Fig 3. Comparison of adipocyte morphology.** (A) Example of white adipocytes from axilla stained with hematoxylin and eosin. (B) Example of the darker BAT-like adipocytes from supraclavicular region stained with hematoxylin and eosin. (C) Adipocyte mean cross sectional area ($\pm$ SEM) of adipose from axilla (Ax) (n = 75 adipocytes from one specimen) versus adipocytes from supraclavicular region (Sc) (n = 75 adipocytes from the same specimen). **** indicates p<0.001 by two-tailed Student's *t*-test.

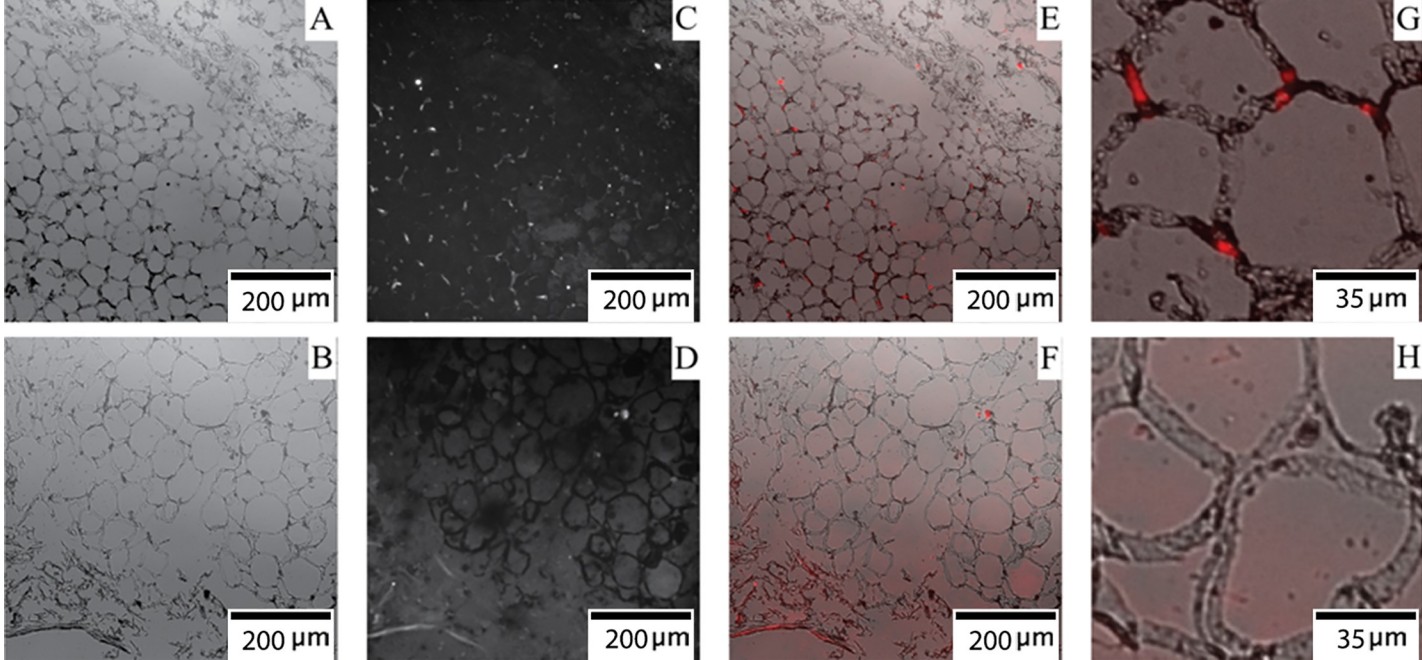

**Fig 4. Tyrosine hydroxylase immunoreactivity in adipose tissue.** BAT-like adipocytes taken from the supraclavicular region imaged with (A) brightfield microscopy, (C) fluorescence microscopy, (E) brightfield and fluorescence merged, (G) brightfield and fluorescence merged and zoomed in. White adipocytes taken from the axilla imaged with (B) brightfield microscopy, (D) fluorescence microscopy, (F) brightfield and fluorescence merged, (H) brightfield and fluorescence merged and zoomed in. D, F and H were captured with 10x longer exposure time than C, E and G.

Finally, to further investigate the nature of the innervation of the BAT-like tissue we set out to test whether there was evidence of sympathetic fibres in the adipose tissue. To do this, we fluorescently immunolabelled tyrosine hydroxylase, a sympathetic nerve marker [39] present in the presumed BAT. Sections taken from the supraclavicular BAT-like tissue revealed small puncta of tyrosine hydroxylase immunoreactive structures. These structures aligned specifically with the inter-adipocyte interstitial space as observed in overlaid brightfield micrographs (Fig 4C, 4E and 4G). The size of these tyrosine hydroxylase immunoreactive structures (approximately 5μm) is in good agreement with the expected diameter of nerve axons when viewed in a transverse section [40], while their presence at adipocyte periphery also matches what would be expected of axons. Given the nerve branch terminated in the adipose tissue, the only possible structures that would be innervated are the adipose tissue itself, or smooth muscle in associated blood vessels. The sympathetic fibres observed are not associated with any vasculature (Fig 4E and 4G), and thus are likely to be innervating the adipose tissue. As a control, we performed the same immunolabelling of confirmed WAT taken from the axilla of the same cadaver. In this WAT, no tyrosine hydroxylase fluorescence was observed with no puncta detected under the same imaging conditions as used for supraclavicular tissue samples. Exposure time was increased in order to bias towards finding immunofluorescent structures, however the limited fluorescence observed did not appear as distinct puncta and showed no correlation with cell peripheral structures (Fig 4D, 4F and 4H).

## Discussion

A depot of adipose tissue in the posterior triangle, that was darker in appearance, was identified in all cadaveric specimens. Consistent with our hypothesis, a nerve branch that has not been previously identified was identified in all specimens. The nerve branch arises from the

cervical plexus and pierces and terminates in adipose tissue in the supraclavicular region. Histological analysis revealed adipocytes from the supraclavicular region did not appear to be classical brown adipocytes, as they contained only a single lipid droplet, rather than the classical multilocular appearance. However, adipocytes from this region, presented with smaller cell size and the tissue generally appeared to represent the intermixture of cell sizes and shapes previously described for human BAT [35]. Presence of sympathetic innervation was also identified in supraclavicular adipose tissue, when compared with axillary adipose tissue from the same specimens, suggesting potential thermogenic capacity. While we cannot conclusively conclude that the supraclavicular adipose tissue is BAT, we have demonstrated sympathetic innervation to the adipose tissue in this region, which has been previously shown to contain BAT.

The supraclavicular adipose tissue analysed in this study appeared with a single intracellular lipid droplet (unilocular), but cells were smaller than adipocytes from the axilla. Whilst BAT is classically described as having multilocular lipid droplets [34] this appearance is typical of cold-exposed rodents [35]. Under thermoneutral conditions rodent BAT presents similarly to human BAT, where adipocytes are unilocular, but comparatively smaller than white adipocytes [35]. Differences in morphology between supraclavicular and axilla adipocytes, that we see, might suggest functional differences. The supraclavicular adipose depot was in a location previously demonstrated, by both PET imaging and histological analysis, to have functional BAT in younger adults (<50 years) [18, 27]. Interestingly, BAT has previously been described as being subcutaneously, or visceral [21], anatomically the supraclavicular adipose tissue is neither immediately deep to the skin (subcutaneous), nor in a visceral cavity (visceral). The supraclavicular BAT-like adipose tissue may need a new location descriptor, given it does not fit existing criteria for subcutaneous or visceral adipose tissue.

While our specimens did not show typical BAT, this is likely to be related to the age of the specimens (mean age 76.7). There is no gold standard for quantification of total body BAT by either anatomical or functional imaging methods. Although there is evidence through functional PET imaging, that the prevalence of active BAT decreases with age [41]. Anatomical and histological analysis has indicated that individuals may possess some thermogenic capacity despite presenting as BAT negative in functional imaging studies [42]. This suggests that functional PET imaging underestimates the prevalence of functional BAT. Although the mean age of the specimens used in this study was 76.7 years, and we could not perform any functional tests on the tissue, the overall BAT-like nature of the supraclavicular adipose tissue depot possibly indicates that perhaps (limited) BAT activity persists in old age.

Here we identified tyrosine hydroxylase immunoreactivity within human supraclavicular adipose tissue. To date, no studies had demonstrated the presence of tyrosine hydroxylase immunoreactive structures in this tissue. Such immunoreactivity is indicative of the presence of sympathetic fibres [39]. This is consistent with previous literature reporting expression of adrenergic receptors by human adipocytes [18, 27]. Further, as this adipose depot mostly lacked smooth-muscle containing vasculature (Fig 4E), these tyrosine hydroxylase immunoreactive structures are most likely to be innervating the adipocytes themselves not associated vasculature. Quantitative or qualitative analysis for a BAT specific gene, such as uncoupling protein 1 (UCP1), would ascertain more conclusively whether the supraclavicular adipose tissue contained brown adipocytes.

Surprisingly, a nerve innervating supraclavicular adipose tissue has not been previously described in authoritative anatomical texts [31]. Here we demonstrate that four of five specimens presented with the same pattern of innervation to supraclavicular adipose tissue. The remaining specimen presented with innervation through a branch of the supraclavicular nerve. Despite this minor difference in the pattern of innervation, our findings corroborate the

hypothesis that supraclavicular adipose tissue is innervated by branches of the cervical plexus. This hypothesis is further supported by the observation that the nerve of interest terminated in the adipose tissue and did not travel beyond to other structures.

The identification and description of this nerve may have therapeutic importance. Researchers and clinicians may be able to access this nerve to monitor sympathetic drive [43] to adipose depots in humans, using nerve recording, as performed in rodents [24]. We speculate that this may aid in assessing the efficacy of potential future pharmacological treatments designed to stimulate BAT activity [24] in obese populations. Additionally, identification of this nerve branch may allow the use of targeted electrical stimulation of the nerve as a means of non-pharmacological treatment for increasing BAT activity to aid weight loss [44]. Although patients with limited functional BAT (obese or elderly) do not appear to be suitable for BAT-related therapies, targeted adrenoceptor-mediated stimulation of adipose tissue has been shown to increase the thermogenic capacity of the tissue in rodents [45]. This may be due to activation of BAT-like adipose reservoirs or recruitment of white adipocytes to BAT-like activity. Recruitment of white adipocytes to perform thermogenesis is known as "beiging", where levels of UCP1 and other thermogenesis related genes are increased within the tissue, thus increasing the efficiency with which heat can be produced [45].

This study clearly identified a nerve terminating in the supraclavicular adipose depot in all specimens. Additionally, tyrosine hydroxylase immunoreactivity characteristic of sympathetic fibres was identified within the adipose tissue. Limitations of this study included sample size (n = 5) and the advanced mean age of the specimens (76.7 years) that made the identification of functional BAT unlikely. Tissue fixation was not appropriate for qualitatively or quantitatively assaying uncoupling protein 1 (UCP1) or any other BAT specific genes. These specimens were originally fixed for teaching purposes in an anatomy department. Additionally, the only specimen to present with a different pattern of innervation did not have the contralateral side available for dissection. Therefore, it was not possible to determine whether the pattern of innervation was bilateral in this specimen.

In summary, a nerve branch from the cervical plexus, was found to pierce and terminate in a depot of adipose in the supraclavicular region. While analysis revealed adipocytes from the supraclavicular region presented with only some brown characteristics, the presence of tyrosine hydroxylase suggests sympathetic innervation and therefore thermogenic capacity. Additionally, this depot of adipose tissue coincides with a location where functional BAT has previously been demonstrated [19, 20]. Identification and description of this nerve may allow researchers and clinicians to exploit it for electroneurography or electrical stimulation for the means of aiding weight loss.

## Materials and methods

This study comprised of two stages; firstly, the investigation of the pattern of the innervation to the supraclavicular adipose tissue, and secondly, a histological examination of the adipose tissue depot and its innervation.

### Cadaveric material

A total of 5 sides from 3 cadavers (mean age 76.7 years) were used in this study. The embalmed cadaveric specimens were obtained from The University of Melbourne body donor program (Table 1). The study was approved by The University of Melbourne Human Research Ethics Committee (Ethics ID: 1953850).

## Anatomical investigation of pattern of innervation

**Dissection procedure.** Each specimen was placed supine on a dissection table, and the clavicle, sternocleidomastoid and trapezius muscle were palpated to identify the region of the posterior triangle (Fig 1). The skin, platysma and deep cervical fascia overlying the posterior triangle were carefully reflected retaining all branches of the cervical plexus. A depot of adipose tissue in the supraclavicular region within the posterior triangle, was then identified in a location consistent with previous functional PET scan studies [27]. All previously described branches of the cervical plexus (e.g. supraclavicular nerves, dorsal scapular nerve, transverse cutaneous nerve of neck, lesser occipital nerve, greater auricular nerve, phrenic nerve) were identified and branches closely related to the supraclavicular depot of adipose tissue were isolated and traced to identify specific branches that terminated in this depot of adipose tissue. Any adipose tissue clearly not associated with the nerve branch was removed. Termination of the nerve branches in adipose tissue was confirmed by freeing the depot and confirming that the nerve branch did not continue beyond it. Photographs were used to document the course of the nerve branches, and tissue samples of the supraclavicular adipose depot and axillary adipose tissue (typically accepted as WAT) were collected for subsequent comparison using histological analysis.

## Histological validation of nature of adipose tissue & innervation

**Immunofluorescence for tyrosine hydroxylase to assess presence of sympathetic fibres.** Samples of adipose tissue were collected from the supraclavicular region, as well as from the axilla for comparison. Tissue underwent a dehydration protocol with increasing concentrations of ethanol, prior to embedding in paraffin wax [46]. Samples were sectioned at 5μm on a rotary microtome (Leitz 1512), followed by immunofluorescence staining for tyrosine hydroxylase [39, 47]. Sectioned tissue was incubated in the anti-tyrosine hydroxylase primary antibody raised in rabbit (Abcam: ab75875) [48] diluted to 1:200 in phosphate buffered saline (PBS) at room temperature overnight. Following incubation with primary antibody, tissue was incubated with Alexafluor 647 goat anti-rabbit (Invitrogen: A27040) diluted to 1:500 in PBS for two hours.

**Haematoxalin and eosin staining to assess morphology.** Hematoxalin and Eosin (H&E) staining was performed with Harris' Hematoxalin (Sigma-Aldrich) and 0.2% eosin (Sigma-Aldrich) in water solution [49].

## Microscopy

Brightfield and fluorescence photomicrographs were made of sections by fluorescence and brightfield microscopy (Olympus IX83 and Photometrics Prime 95B). Fiji image processing software was used to analyse photomicrographs [50]. Both qualitative and quantitative methods were used to compare the morphology of adipose tissue taken from the supraclavicular region with WAT from the axilla. Qualitative analysis included the presence or absence of tyrosine hydroxylase, indicating the presence or absence of sympathetic fibres respectively [39].

Quantitative analysis of adipose cell morphology was performed by a blinded assessor. Measurements of cell cross sectional area were calculated from 75 adipocytes, a similar number to that reported in a previous study, [51] for each region (supraclavicular n = 75, axilla n = 75). These cells were from six sections, from three separate blocks of tissue from a single cadaver specimen. A minimum of 10 adjacent, intact adipocytes were identified per section for analysis. A students t-test was used to check for statistical difference in cell morphology (cross sectional area), with p-values <0.05 considered significant.

## Supporting information

**S1 File.**
(PDF)

## Acknowledgments

The University of Melbourne body donors and Dr Simon Murray, Department of Anatomy & Neuroscience, University of Melbourne for facilitating the ethical approval for this project.

## Author Contributions

**Conceptualization:** Will Sievers, Joseph A. Rathner, Rodney A. Green, Christine Kettle, Helen R. Irving, Anita Zacharias.

**Data curation:** Will Sievers, Rodney A. Green, Donna R. Whelan, Richard G. D. Fernandez, Anita Zacharias.

**Formal analysis:** Will Sievers, Rodney A. Green, Donna R. Whelan, Anita Zacharias.

**Investigation:** Will Sievers, Rodney A. Green, Christine Kettle, Helen R. Irving, Donna R. Whelan, Richard G. D. Fernandez, Anita Zacharias.

**Methodology:** Will Sievers, Joseph A. Rathner, Rodney A. Green, Donna R. Whelan, Richard G. D. Fernandez, Anita Zacharias.

**Project administration:** Will Sievers, Joseph A. Rathner, Rodney A. Green, Christine Kettle, Helen R. Irving, Donna R. Whelan, Anita Zacharias.

**Resources:** Will Sievers, Richard G. D. Fernandez, Anita Zacharias.

**Software:** Will Sievers, Donna R. Whelan.

**Supervision:** Joseph A. Rathner, Rodney A. Green, Christine Kettle, Helen R. Irving, Anita Zacharias.

**Visualization:** Will Sievers, Helen R. Irving, Donna R. Whelan, Anita Zacharias.

**Writing – original draft:** Will Sievers.

**Writing – review & editing:** Will Sievers, Joseph A. Rathner, Rodney A. Green, Christine Kettle, Helen R. Irving, Donna R. Whelan, Richard G. D. Fernandez, Anita Zacharias.

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
