## [Decision Letter · Decision Letter 0]

28 May 2020

PONE-D-20-13612

Innervation of supraclavicular adipose tissue: A human cadaveric study

PLOS ONE

Dear Dr. Sievers,

Thank you for submitting your manuscript to PLOS ONE. After careful consideration, we feel that it has merit but does not fully meet PLOS ONE’s publication criteria as it currently stands. Therefore, we invite you to submit a revised version of the manuscript that addresses the points raised during the review process.

We look forward to receiving your revised manuscript.

Kind regards,

Jo Edward Lewis, Ph.D

Academic Editor

PLOS ONE

Journal Requirements:

Additional Editor Comments (if provided):

The introduction could be slightly more nuanced for example when discussing weight loss via caloric restriction (line 50-52). A reference to weight loss surgeries (e.g. vertical sleeve gastrectomy, Roux-en-Y bypass) and pharmacological interventions (e.g. Semaglutide) would be welcomed. These would result in an additional discussion point – the effect of surgery and pharmacological intervention on BAT.

Similarly, line 56-59, pharmacological activation of BAT in humans have failed in clinical trials due to their B1 and B2 adrenoreceptor mediated cardiovascular and muscular events.

Line 111 five dissections – please clarify this section, was it 5 cadavers? For 5 samples from 3 cadavers? Please provide justification for missing specimen i.e. if 3 cadavers, why not 6 specimens (left and right)? NOTE: I see this is provided in the discussion, please add to the results section.

Please address these comments and those of the reviewers.

Reviewers' comments:

Reviewer's Responses to Questions

**Comments to the Author**

1. Is the manuscript technically sound, and do the data support the conclusions?

Reviewer #1: Yes

Reviewer #2: Partly

2. Has the statistical analysis been performed appropriately and rigorously? 

Reviewer #1: N/A

Reviewer #2: No

3. Have the authors made all data underlying the findings in their manuscript fully available?

Reviewer #1: Yes

Reviewer #2: No

4. Is the manuscript presented in an intelligible fashion and written in standard English?

Reviewer #1: Yes

Reviewer #2: Yes

5. Review Comments to the Author

Reviewer #1: The manuscript by Sievers and colleagues contains novel data on the innervation of what is most probably brown adipose tissue in adult humans. As acknowledged by the authors, the study is limited by its small size and by the fact that the samples were from elderly people where active brown fat has not been reported. However, the authors argue convincingly that the innervation they identify is both unique and relevant, and indeed could be a novel means to activate brown fat in the future. As has been noted in various studies, even apparently inactive BAT can be reactivated by suitable stimuli.

I only have minor comments.

In rodents, it is reported that the sympathetic nerves synapse with the adipocytes as boutons en passant. Do the authors have any possibility to evaluate this in their samples?

Also, they should note that white adipose tissue is also sympathetically innervated, although far less densely.

They also state that the physical innervation in adult human BAT as not been studied, which I believe is correct. However, there are several reports of e.g. PET studies demonstrating norepinephrine transporters, uptake of agonist analogs e.g. C-HED etc. that do indicate that the tissue is, as expected, sympathetically innervated, although not from where. These studies should be mentioned and referenced.

I was personally unable to clearly see the color difference that the authors claim between the areas marked 5 and 8 in Fig. 2. Can they describe it better or enhance it?

l. 59. There are at least two publications that show that BAT correlates positively with obesity – it is increased as obesity increases in an “attempt” to decrease fat storage. This is the same as in high-fat diet fed rodents. However, it looks to be inactive as it accumulates large amounts of triglyceride.

l. 62 – 64. Reformulations needed. BAT can contribute to the heat required to create a fever but even without BAT, an organism will generate a fever if the brain requires this. BAT does not influence basal metabolic rate but can increase resting metabolic rate.

Reviewer #2: This is a potentially important paper. However, the authors need to correct their terminology throughout re the number of subjects examined which is of course 3 with duplicate measures for 2 – i.e. left and right sides NOT 5. Representative anatomical images of all subjects are thus required. The statistical analysis on the histology is similarly not valid. The n is the number of subjects not the number of samples measured. Figure 3C in particular needs correcting.

6. PLOS authors have the option to publish the peer review history of their article (what does this mean?). If published, this will include your full peer review and any attached files.

Reviewer #1: No

Reviewer #2: Yes: M E Symonds

---

## [Author Response · Author response to Decision Letter 0]

28 Jun 2020

Please note: Not all formatting has tranferred into this text-box. Please refer to attached word document "Rebuttal_Letter.docx". 

Manuscript Number: PONE-D-20-13612

Title: Innervation of supraclavicular adipose tissue: A human cadaveric study

Response to reviewer’s comments:

We thank the reviewers for the responses and comments. We have responded to all comments made by the reviewers. Please note that the page and line numbers reflect the revised version (document with tracked changes) of the manuscript. Additionally, the reference numbers differ from those in the manuscript. All references included in the manuscript have been highlighted. 

Additional Editor Comments:  Comment 1: The introduction could be slightly more nuanced for example when discussing weight loss via caloric restriction (line 50-52). A reference to weight loss surgeries (e.g. vertical sleeve gastrectomy, Roux-en-Y bypass) and pharmacological interventions (e.g. Semaglutide) would be welcomed. These would result in an additional discussion point – the effect of surgery and pharmacological intervention on BAT. Similarly, line 56-59, pharmacological activation of BAT in humans have failed in clinical trials due to their B1 and B2 adrenoreceptor mediated cardiovascular and muscular events. 

Response: 

An additional two sentences have been added to the end of the first introductory paragraph (lines 52-56), to address these points.

“Beyond lifestyle intervention, bariatric surgery is invasive and presents risks of complications such as pulmonary embolism, enteric leak or gastrointestinal bleeding (1). While pharmacological intervention is also an option, these drugs (i.e. glucagon-like peptide-1 receptor agonists, or adrenergic receptor agonists) typically have gastrointestinal or cardiovascular side effects (2, 3).”

 Comment 2: Line 111 five dissections – please clarify this section, was it 5 cadavers? For 5 samples from 3 cadavers? Please provide justification for missing specimen i.e. if 3 cadavers, why not 6 specimens (left and right)? NOTE: I see this is provided in the discussion, please add to the results section. 

Response:

Anatomical variation is not uncommon between right and left sides, so it is fairly common to cite the number of dissections completed, nevertheless, the manuscript has been modified according to your suggestion (line 124-126). Terminology has also been modified, in order to refer to the number of dissections as five, and number of specimens as three (lines 125, 137, 140, 143, 144). 

“In order to investigate the innervation of adipose tissue within the supraclavicular region, five dissections of three adult cadavers were performed. A sixth dissection could not be performed as the right side of specimen 1 had already been dissected for teaching purposes. To begin the dissection, the region of the posterior triangle was identified by locating the sternocleidomastoid muscle, collarbone (clavicle) and trapezius muscle (Figure 1). The skin, muscle and connective tissue overlying the posterior triangle were removed retaining all nerve branches of the cervical plexus (Figure 2). A depot of adipose tissue in the supraclavicular region within the posterior triangle, a location consistent with previous functional PET scan studies (4), was tentatively identified as BAT based on coloration (5). This is clearly shown in Figure 2B by comparing WAT (structure 8) with supraclavicular depot of adipose tissue (structure 5). All previously described branches of the cervical plexus were identified and branches closely related to the supraclavicular depot of adipose tissue were isolated and followed to identify specific branches that terminated in adipose tissue. In all five dissections, branches of the cervical plexus were identified in the posterior triangle, including the relatively superficial supraclavicular branches that have been previously described as innervating skin over the posterior triangle and more inferiorly onto the chest and shoulder regions. A single nerve within each of the three specimens was found to terminate in the visibly darker adipose tissue, and therefore possibly BAT. Interestingly, two distinct patterns of innervation to this adipose tissue were identified (Table 1). In four of the five dissections, the nerve emerged as an independent branch of the cervical plexus from the ventral ramus of third cervical spinal nerve (Figure 2a). In the remaining dissection, the nerve emerged as a branch of the supraclavicular nerve (Figure 2b). This marks the first time innervation to supraclavicular adipose tissue has been reported in humans.”

 Please address these comments and those of the reviewers.   Reviewers' comments:  Reviewer's Responses to Questions  Comments to the Author  Comment 1. Is the manuscript technically sound, and do the data support the conclusions?  The manuscript must describe a technically sound piece of scientific research with data that supports the conclusions. Experiments must have been conducted rigorously, with appropriate controls, replication, and sample sizes. The conclusions must be drawn appropriately based on the data presented.   Reviewer #1: Yes  Reviewer #2: Partly

Response:

See response to Reviewer #2, comments 2 and 3 below.

 Comment 2. Has the statistical analysis been performed appropriately and rigorously?   Reviewer #1: N/A  Reviewer #2: No

Response:

See response to Reviewer #2, comment 3 below.

 Comment 3. Have the authors made all data underlying the findings in their manuscript fully available?  The PLOS Data policy requires authors to make all data underlying the findings described in their manuscript fully available without restriction, with rare exception (please refer to the Data Availability Statement in the manuscript PDF file). The data should be provided as part of the manuscript or its supporting information, or deposited to a public repository. For example, in addition to summary statistics, the data points behind means, medians and variance measures should be available. If there are restrictions on publicly sharing data—e.g. participant privacy or use of data from a third party—those must be specified.  Reviewer #1: Yes  Reviewer #2: No

Response:

As requested, all raw data for adipocyte cross sectional area (S1 Table) and images of all dissections (S1 Fig) have been provided as supporting information.

 Comment 4. Is the manuscript presented in an intelligible fashion and written in standard English?  PLOS ONE does not copyedit accepted manuscripts, so the language in submitted articles must be clear, correct, and unambiguous. Any typographical or grammatical errors should be corrected at revision, so please note any specific errors here.  Reviewer #1: Yes  Reviewer #2: Yes

Response:

No response required 

 5. Review Comments to the Author  Please use the space provided to explain your answers to the questions above. You may also include additional comments for the author, including concerns about dual publication, research ethics, or publication ethics. (Please upload your review as an attachment if it exceeds 20,000 characters)  Reviewer #1: The manuscript by Sievers and colleagues contains novel data on the innervation of what is most probably brown adipose tissue in adult humans. As acknowledged by the authors, the study is limited by its small size and by the fact that the samples were from elderly people where active brown fat has not been reported. However, the authors argue convincingly that the innervation they identify is both unique and relevant, and indeed could be a novel means to activate brown fat in the future. As has been noted in various studies, even apparently inactive BAT can be reactivated by suitable stimuli. I only have minor comments.

 Comment 1: In rodents, it is reported that the sympathetic nerves synapse with the adipocytes as boutons en passant. Do the authors have any possibility to evaluate this in their samples? Also, they should note that white adipose tissue is also sympathetically innervated, although far less densely.

Response:

It is a very interesting suggestion, and we would like to pursue this in the future, however at this point in time we simply do not have appropriate tissue available to conduct the additional immunohistochemistry. We only had access to fixed specimens for teaching rather than research purposes. No change to the manuscript as this is already acknowledged in the discussion (lines 306-309).

 Comment 2: They also state that the physical innervation in adult human BAT as not been studied, which I believe is correct. However, there are several reports of e.g. PET studies demonstrating norepinephrine transporters, uptake of agonist analogs e.g. C-HED etc. that do indicate that the tissue is, as expected, sympathetically innervated, although not from where. These studies should be mentioned and referenced.

Response:

As suggested, we have amended the first two sentences of the fifth introduction paragraph (lines 97-100); 

“There is evidence to suggest that human BAT expresses adrenoceptors (4, 6), and possesses norepinephrine transporters (7, 8), providing some indication that human BAT is under sympathetic regulation. However, no specific peripheral nerve responsible for carrying innervation to BAT has been identified to date.”

 Comment 3: I was personally unable to clearly see the color difference that the authors claim between the areas marked 5 and 8 in Fig. 2. Can they describe it better or enhance it?

Response: 

We have added a label for structure 8, in Figure 2B. The contrast in colour between structures 8 and 5, is more apparent in Figure 2B. The manuscript has also been modified to refer to figure 2B specifically (line 133). 

“This is clearly shown in Figure 2B by comparing WAT (structure 8) with supraclavicular depot of adipose tissue (structure 5).”

 Comment 4: l. 59. There are at least two publications that show that BAT correlates positively with obesity – it is increased as obesity increases in an “attempt” to decrease fat storage. This is the same as in high-fat diet fed rodents. However, it looks to be inactive as it accumulates large amounts of triglyceride.

Response: 

The suggested publications provided by the journal have been incorporated and they support the conclusion that there is a negative correlation between BAT thermogenic capacity and obesity. This is consistent with our original statement that BAT mass is inversely correlated with obesity. This has been incorporated into the manuscript (lines 62-64).

“While active BAT has been reported in obese individuals (9), studies have consistently observed in humans that obesity is inversely correlated with BAT activity (10-12) and BAT volume (13).” 

 Comment 5: l. 62 – 64. Reformulations needed. BAT can contribute to the heat required to create a fever but even without BAT, an organism will generate a fever if the brain requires this. BAT does not influence basal metabolic rate but can increase resting metabolic rate.

Response:

The manuscript has been modified as per the reviewer suggestion. Fever has been removed as a manifestation of BAT thermogenesis (lines 64-65), and “basal metabolic rate” has been changed to “resting metabolic rate” (lines 69-71).

“This can manifest in two ways: BAT can help maintain optimum body temperature in cold conditions, or increase resting metabolic rate in order to maintain a healthy body-weight (14-16).”

Reviewer #2: This is a potentially important paper. 

Comment 1: However, the authors need to correct their terminology throughout re the number of subjects examined which is of course 3 with duplicate measures for 2 – i.e. left and right sides NOT 5.

Response: 

Anatomical variation is not uncommon between right and left sides, so it is common practice to both, publish with small numbers of specimens, and to cite the number of dissections completed (17, 18), nevertheless, terminology has been changed in the following parts of the manuscript to identify both the number of cadavers/specimens and dissections: Lines 124-146, 302-303. 

“In order to investigate the innervation of adipose tissue within the supraclavicular region, five dissections of three adult cadavers were performed. A sixth dissection could not be performed as the right side of specimen 1 had already been dissected for teaching purposes. To begin the dissection, the region of the posterior triangle was identified by locating the sternocleidomastoid muscle, collarbone (clavicle) and trapezius muscle (Figure 1). The skin, muscle and connective tissue overlying the posterior triangle were removed retaining all nerve branches of the cervical plexus (Figure 2). A depot of adipose tissue in the supraclavicular region within the posterior triangle, a location consistent with previous functional PET scan studies (4), was tentatively identified as BAT based on coloration (5). This is clearly shown in Figure 2B by comparing WAT (structure 8) with supraclavicular depot of adipose tissue (structure 5). All previously described branches of the cervical plexus were identified and branches closely related to the supraclavicular depot of adipose tissue were isolated and followed to identify specific branches that terminated in adipose tissue. In all five dissections, branches of the cervical plexus were identified in the posterior triangle, including the relatively superficial supraclavicular branches that have been previously described as innervating skin over the posterior triangle and more inferiorly onto the chest and shoulder regions. A single nerve within each of the three specimens was found to terminate in the visibly darker adipose tissue, and therefore possibly BAT. Interestingly, two distinct patterns of innervation to this adipose tissue were identified (Table 1). In four of the five dissections, the nerve emerged as an independent branch of the cervical plexus from the ventral ramus of third cervical spinal nerve (Figure 2a). In the remaining dissection, the nerve emerged as a branch of the supraclavicular nerve (Figure 2b). This marks the first time innervation to supraclavicular adipose tissue has been reported in humans.”

“This study clearly identified a nerve terminating in the supraclavicular adipose depot in all specimens.”

Comment 2: Representative anatomical images of all subjects are thus required.

Response:

Images of all dissections showing the innervation of the adipose tissue depot have been added as supporting information (S1 Fig).

Comment 3: The statistical analysis on the histology is similarly not valid. The n is the number of subjects not the number of samples measured. Figure 3C in particular needs correcting.

Response:

The authors agree that comparison of single adipocytes from each specimen (implied where the n=3 refers to the number of specimens) would not be statistically valid. Therefore, significant blocks of adipose tissue from the axilla (WAT) and the supraclavicular region were taken from the same specimen for detailed analysis. Thus the n for the statistical comparison of cell morphology is the number of adipocytes compared from each region. n is the sample size, and as we are taking samples from a population of adipocytes, n = 75. Studies have reported that intra-individual site-to-site variability in adipocyte size is greater than variability of the same site between individuals (19-21). Therefore, samples from one specimen should be sufficient to establish whether adipocytes from these two sites have differing morphology. This justification has been added to the results section (lines 179-181). The legend for Figure 3C (line 193-195) has also been modified to reflect the number of subjects, as well as the sample size of adipocytes. 

“Samples from a single specimen were used as studies have reported that intra-individual site-to-site variability in adipocyte size is greater than variability of the same site between individuals (19-21).” 

“(C) Adipocyte mean cross sectional area (± SEM) of adipose from axilla (Ax) (n = 75 adipocytes from one specimen) versus adipocytes from supraclavicular region (Sc) (n = 75 adipocytes from one specimen).”

\f

References

1. Jameson JL, De Groot LJ, de Kretser DM, Giudice LC, Grossman AB, Melmed S, et al. Endocrinology: Adult and Pediatric. 7th Edition ed. Philadelphia PA USA: Elsevier; 2016.

2. Vilsbøll T, Christensen M, Junker AE, Knop FK, Gluud LL. Effects of glucagon-like peptide-1 receptor agonists on weight loss: systematic review and meta-analyses of randomised controlled trials. BMJ. 2012;344:d7771.

3. Chen KY, Brychta RJ, Abdul Sater Z, Cassimatis TM, Cero C, Fletcher LA, et al. Opportunities and challenges in the therapeutic activation of human energy expenditure and thermogenesis to manage obesity. J Biol Chem. 2020;295(7):1926-42.

4. Cypess AM, Weiner LS, Roberts-Toler C, Franquet Elia E, Kessler SH, Kahn PA, et al. Activation of human brown adipose tissue by a beta3-adrenergic receptor agonist. Cell metabolism. 2015;21(1):33-8.

5. Lee P, Swarbrick MM, Ho KKY. Brown Adipose Tissue in Adult Humans: A Metabolic Renaissance. Endocrine Reviews. 2013;34(3):413-38.

6. Virtanen KA, Lidell ME, Orava J, Heglind M, Westergren R, Niemi T, et al. Functional brown adipose tissue in healthy adults. The New England journal of medicine. 2009;360(15):1518-25.

7. Hwang JJ, Yeckel CW, Gallezot JD, Aguiar RB, Ersahin D, Gao H, et al. Imaging human brown adipose tissue under room temperature conditions with (11)C-MRB, a selective norepinephrine transporter PET ligand. Metabolism. 2015;64(6):747-55.

8. Muzik O, Mangner TJ, Leonard WR, Kumar A, Granneman JG. Sympathetic Innervation of Cold-Activated Brown and White Fat in Lean Young Adults. Journal of nuclear medicine : official publication, Society of Nuclear Medicine. 2017;58(5):799-806.

9. Mihalopoulos NL, Yap JT, Beardmore B, Holubkov R, Nanjee MN, Hoffman JM. Cold-Activated Brown Adipose Tissue is Associated with Less Cardiometabolic Dysfunction in Young Adults with Obesity. Obesity. 2020;28(5):916-23.

10. Loh RKC, Kingwell BA, Carey AL. Human brown adipose tissue as a target for obesity management; beyond cold-induced thermogenesis. Obesity reviews. 2017;18(11):1227-42.

11. Vijgen GH, Bouvy ND, Teule GJ, Brans B, Schrauwen P, van Marken Lichtenbelt WD. Brown adipose tissue in morbidly obese subjects. PloS one. 2011;6(2):e17247.

12. Madden CJ, Morrison SF. A high-fat diet impairs cooling-evoked brown adipose tissue activation via a vagal afferent mechanism. American Journal of Physiology-Endocrinology and Metabolism. 2016;311(2):E287-E92.

13. Leitner BP, Huang S, Brychta RJ, Duckworth CJ, Baskin AS, McGehee S, et al. Mapping of human brown adipose tissue in lean and obese young men. Proceedings of the National Academy of Sciences. 2017;114(32):8649-54.

14. Kajimura S, Saito M. A new era in brown adipose tissue biology: molecular control of brown fat development and energy homeostasis. Annual review of physiology. 2014;76:225-49.

15. Sievers W, Rathner JA, Kettle C, Zacharias A, Irving HR, Green RA. The capacity for oestrogen to influence obesity through brown adipose tissue thermogenesis in animal models: A systematic review and meta-analysis. Obesity science & practice. 2019;5(6):592-602.

16. Spiegelman BM, Flier JS. Obesity and the regulation of energy balance. Cell. 2001;104(4):531-43.

17. Rathi S, Zacharias A, Green RA. Verification of a standardized method for inserting intramuscular electromyography electrodes into teres minor using ultrasound. Clinical anatomy (New York, NY). 2015;28(6):780-5.

18. Semciw AI, Green RA, Pizzari T, Briggs C. Verification of a standardized method for inserting intramuscular EMG electrodes into uniquely oriented segments of gluteus minimus and gluteus medius. Clinical anatomy (New York, NY). 2013;26(2):244-52.

19. Salans LB, Horton ES, Sims EA. Experimental obesity in man: cellular character of the adipose tissue. J Clin Invest. 1971;50(5):1005-11.

20. McLaughlin T, Lamendola C, Coghlan N, Liu TC, Lerner K, Sherman A, et al. Subcutaneous adipose cell size and distribution: relationship to insulin resistance and body fat. Obesity (Silver Spring, Md). 2014;22(3):673-80.

21. Tchoukalova YD, Votruba SB, Tchkonia T, Giorgadze N, Kirkland JL, Jensen MD. Regional differences in cellular mechanisms of adipose tissue gain with overfeeding. Proceedings of the National Academy of Sciences. 2010;107(42):18226-31.

---

## [Editor Report · Decision Letter 1]

6 Jul 2020

Innervation of supraclavicular adipose tissue: A human cadaveric study

PONE-D-20-13612R1

Dear Dr. Sievers,

We’re pleased to inform you that your manuscript has been judged scientifically suitable for publication and will be formally accepted for publication once it meets all outstanding technical requirements.

Kind regards,

Jo Edward Lewis, Ph.D

Academic Editor

PLOS ONE
---

## [Editor Report · Acceptance letter]

13 Jul 2020

PONE-D-20-13612R1 

Innervation of supraclavicular adipose tissue: A human cadaveric study 

Dear Dr. Sievers:

I'm pleased to inform you that your manuscript has been deemed suitable for publication in PLOS ONE. Congratulations! Your manuscript is now with our production department. 

Kind regards, 

on behalf of

Dr. Jo Edward Lewis 

Academic Editor

PLOS ONE